# MDABP: A Novel Approach to Detect Cross-Architecture IoT Malware Based on PaaS

**DOI:** 10.3390/s23063060

**Published:** 2023-03-13

**Authors:** Yang Zhao, Alifu Kuerban

**Affiliations:** The College of Software, Xinjiang University, Urumqi 830046, China

**Keywords:** IoT malware, cross-architecture, dynamic analysis, PaaS, system call

## Abstract

With the development of internet technology, the Internet of Things (IoT) has been widely used in several aspects of human life. However, IoT devices are becoming more vulnerable to malware attacks due to their limited computational resources and the manufacturers’ inability to update the firmware on time. As IoT devices are increasing rapidly, their security must classify malicious software accurately; however, current IoT malware classification methods cannot detect cross-architecture IoT malware using system calls in a particular operating system as the only class of dynamic features. To address these issues, this paper proposes an IoT malware detection approach based on PaaS (Platform as a Service), which detects cross-architecture IoT malware by intercepting system calls generated by virtual machines in the host operating system acting as dynamic features and using the K Nearest Neighbors (KNN) classification model. A comprehensive evaluation using a 1719 sample dataset containing ARM and X86-32 architectures demonstrated that MDABP achieves 97.18% average accuracy and a 99.01% recall rate in detecting samples in an Executable and Linkable Format (ELF). Compared with the best cross-architecture detection method that uses network traffic as a unique type of dynamic feature with an accuracy of 94.5%, practical results reveal that our method uses fewer features and has higher accuracy.

## 1. Introduction

The Internet of Things (IoT) has been widely used in everyday life since the development of internet technology, and there is a growing number of IoT devices actively used as smart home appliances, embedded systems, and other devices. It is expected that by 2030 [1], the number of IoT devices in use, globally, will reach 50 billion. However, the IoT devices’ limited computational performance and storage capability make traditional security mechanisms impractical [2]. Furthermore, the lack of research and development combined with the poor financial investment in information security by device manufacturers leads to an inherent flaw in the ability of IoT devices to withstand cyberattacks. This has resulted in numerous cyberattacks against IoT devices. For instance, Mirai launched a Distributed Denial of Service (DDoS) attack against an American domain name for a resolution server provider and controlled many IoT devices. This caused a widespread internet outage in eastern America and prevented websites such as Twitter and Amazon from being accessed [3]. This incident indicates that although IoT devices bring great economic benefits, they present significant safety challenges.

Current IoT malware analysis methods are static, dynamic, and hybrid based on the different feature types used. Most static analysis methods employ features from a binary file, such as strings and operation codes (opcodes). Moreover, the dynamic analysis method exploits the behavioral characteristics while monitoring the executed file, such as system calls generated by the interaction with the system. Besides, hybrid analysis methods comprise features of both static and dynamic processes.

Rapid advances in software development techniques allow the source code of malware to be compiled into different executable programs tailored to different central processing unit (CPU) architectures via toolchains [4]. In addition to accelerating the spread of IoT malware, this also means that malware detection methods [5,6] based on a single CPU architecture do not meet the analysis requirements any longer. Given that the opcodes, instruction sets, and other characteristics of the executable files compiled on various CPU architectures differ [7], the static analysis method based on a single CPU architecture uses features that do not contain features of samples compiled based on other CPU architectures. Dynamic analysis methods based on a single CPU architecture cannot obtain features because they cannot run samples compiled based on other CPU architectures. Because cross-architecture malware detection methods can detect IoT malware compiled based on different CPU architectures. Therefore, cross-architecture malware detection methods are better suited for future needs than those based on a single CPU architecture.

A drawback of existing static cross-architecture analysis methods is their inability to obtain valuable features to determine if a sample is malicious because it has been encrypted or obfuscated. Furthermore, if malware authors use packaging techniques correctly, they can completely block any attempts to apply static analysis to malware, forcing researchers to discard samples that could not be disassembled. However, discarded samples may have various characteristics, decreasing the classifier’s ability to detect unknown malware. Unlike static analysis, dynamic analysis can detect malware by simply running the sample and obtaining information about its runtime behavior. Consequently, dynamic analysis can extract key behavioral characteristics even if samples are packed or obfuscated.

Regarding hybrid features, Carrillo-Mondéjar et al. [8] clustered unknown malware with known malware through similarity calculation, exploiting “unique syscalls” as a dynamic feature. However, their dataset includes samples compiled from five different CPU architectures, where the system calls may vary depending on the operating system (OS), and developers of the same OS add or change the corresponding system calls to support the operating system executing on the utilized CPU architecture. Therefore, if all system calls in a specific operating system version developed based on a single CPU architecture are counted as a single set, then five sets of system calls may be involved in the similarity calculation. After deleting the same-named system calls, the number of system calls for the computation can be much larger than a single set of system calls. Nevertheless, approaches based on unique features increase the number of features and the diversity of feature sources, thus increasing the complexity of the feature analysis operation. Given that existing research methods are similar to detection methods [8], no current dynamic analysis method uses a set of system calls as a feature to detect cross-architecture IoT malware. Specifically, there is no method for detecting cross-architecture IoT malware by solely relying on system calls in a single Linux operating system built on a single CPU architecture as dynamic features.

IoTPOT [9] was proposed in 2016, comprising a sandbox named IoTBOX that supported eight CPU architectures. The IoTBOX was constructed using the emulation software QEMU and contained multiple virtual machines (VMs) created with the Linux operating system. The authors used a network bridge to connect these VMs to the host network and characterized the malware family by running the DDoS attack in terms of network traffic over individual ports.

Kernel-based Virtual Machine (KVM) has been extensively studied through the IoTBOX construction process. In this virtualization technology, each VM created on the host is equivalent to a QEMU process, which corresponds to a standard process on the host. Moreover, each VM’s virtual processor is equivalent to a thread of the standard process, i.e., the virtual machine can be treated as a translator that translates instructions from a CPU architecture dissimilar from the host’s instructions into instructions from the CPU that the host is using. Thus, when the sample is run on a virtual machine, the system calls that the VM dispatches to the host are the sample’s behavioral features after being automatically mapped and normalized. Therefore, we developed an IoT malware detection approach based on Platform as a Service (MDABP) that exploits the above principle. Specifically, MDABP uses system calls in the Ubuntu OS developed on X86-64 processors with features that detect malware compiled from the IoT on different CPU architectures. Our method belongs to the dynamic analysis class, which obtains features by running samples and thus avoids the problem where static analysis requires discarding samples that cannot be decompiled. Collecting dynamic features based on the PaaS (Platform as a Service) model reduces the number of system calls used as features.

The developed MDABP method is substantially different from the malware detection methods based on VM introspection (VMI) [10,11]. This is because, in VMI-based schemes, the interrupt instructions that obtain the system call sequences generated by processes inside the VM are considered dynamic features. Opposing VMI, our method uses system calls generated by the VM in the host operating system as dynamic features to detect malware samples. To our knowledge, the proposed MDABP is the first cross-architecture dynamic analysis method that uses a single set of system calls. In this study, we created a dataset consisting of 442 benign samples and 1277 malicious samples, covering CPU architectures that include X86-32 and Advanced RISC machines (ARM). Moreover, the accuracy of MDABP was evaluated on this dataset, attaining 97.18% average accuracy, employing 146 features (such as “ppoll” and “futex”) for distinguishing benign and malicious samples.

The main contributions of this paper are as follows:(1)A novel cross-architecture dynamic analysis method named MDABP is proposed. MDABP is the first cross-architecture dynamic analysis method using a single system call set. This method uses system calls from an Ubuntu OS based on a single CPU architecture as dynamic features to detect IoT malware from multiple CPU architectures.(2)We comprehensively evaluate the performance of MDABP through experiments and compare it with existing analysis methods to establish its efficiency.

The remainder of this paper is organized as follows: Section 2 reviews the related work on cross-architecture IoT malware detection. Section 3 details the approach based on PaaS for cross-architecture IoT malware detection. Section 4 presents the experimental evaluation, results, and analysis, and finally, Section 5 concludes this work.

## 2. Related Works

This section reviews the research related to cross-architecture approaches for IoT malware detection. Existing malware detection methods are divided into static and dynamic analysis methods based on the features they use.

### 2.1. Static Feature-Based Methods

A comprehensive survey of static detection methods for IoT malware was conducted by Ngo et al. [4]. The authors objectively evaluated the existing static analysis methods using the same dataset and experimental setup as all other competitor methods. Raju et al. [12] comprehensively surveyed cross-architecture malware detection methods and focused on static cross-architecture detection methods. Besides, Tian et al. [10] discuss practical challenges to existing research.

The static analysis methods that use strings [7,13,14,15,16,17] and byte sequences [18,19,20] as features extract a large number of unique features, imposing a computationally intensive and time-consuming detection process. However, reducing the number of features through preprocessing discards valuable data. An alternative solution is employing a static detection method [21,22] that uses control flow graphs as features, which affords high accuracy but is time-consuming to compute. Image-based static analysis [23,24,25,26] often requires complex models with tens of thousands of training parameters. Nevertheless, such approaches may lose accuracy when using obfuscation and encryption techniques to process samples. It should be noted that since different CPU architectures have different instruction sets, the opcodes are also different. Therefore, detecting malware samples across architectures requires considering the opcodes in each CPU architecture, leading to a computationally intensive process [27,28,29,30,31,32,33,34,35,36,37,38].

### 2.2. Dynamic Feature-Based Methods

Dinakarrao et al. [39] suggested a detection method to execute malware using hardware performance counter (HPC) values as features. HPC attained a runtime malware detection accuracy of 92.21%. Moreover, Li et al. [40] proposed detecting malware by extracting identifiable power consumption features from side-channel data. Based on the experimental results, they achieved 99.86% detection accuracy. Pham et al. [41] introduced a method to identify malware threats against IoT devices using electromagnetic information from side-channel analysis techniques. Their method detected malware with 99.89% accuracy. Catuogno et al. [42] collected CPU utilization and network bandwidth as features extracted from virtualized containerized systems to detect the presence of malware. However, hardware performance metrics-based testing involves expensive test equipment. For example, the researcher needs to purchase hardware such as a CPU. Furthermore, the testing methods based on virtualization systems are limited by the CPU architectures supported by the virtualization software.

Besides, Yu et al. [43] suggested a home security smart analytics system utilizing a home router that classifies and identifies malware based on protocol, address, and main content of packets as features. Alrubayyi et al. [44] designed an algorithm based on forward and backward selection techniques. The algorithm used 16-bit strings generated from target port-based network traffic as features and matched them with existing features to detect malware. Kumar et al. [45] proposed a modular scheme called EDIMA for IoT malware network activity detection. This scheme extracted feature vectors from the packets captured by the gateway and classified them. They collected 60 network traffic sessions of 15 min duration for both benign and malicious classes through their testbed. Then they applied three classification algorithms to the traffic sessions rather than individual packets because per-packet classification is computationally much more costly and does not yield any significant benefits. Palla et al. [46] detected Mirai viruses in IoT devices using neural networks to highlight packets in network traffic. Their scheme used Artificial Neural Networks (ANN) to compute the accuracy. ANN learns non-linear functions using the activation function, with the neuron comprising hidden layers to prevent non-linearity and over-fitting. Moreover, they considered the dataset of Damini Doorbell and performed the training based on the data allocation to each stage of the training set. The best accuracy of their method reached 92.9% after experimental validation. Bendiab et al. [47] proposed an approach to quickly detect and classify malware using deep learning and visual representation, where network packets were employed as features. They first captured the network traffic and then converted it into a 2D image with a size of 784 (1024 * 256) bytes. Next, the Residual Neural Network (ResNet50) was used to analyze the produced images against its in-depth training. The ResNet-50 was 50 layers deep and had over 23 million trainable parameters. Their detection accuracy reached 94.50%. Guizani et al. [48] introduced a detection method based on the temporal features of network traffic and recurrent neural networks with long short-term memory (RNN-LSTM) for classification. Their method was experimentally verified and attained 93% accuracy. Muthanna et al. [49] proposed a hybrid intelligent framework for software-defined networking support, leveraging the Cuda Long Short Term Memory Gated Recurrent Unit (cuLSTMGRU) for effective threat detection in IoT environments. Praveena et al. [50] proposed an intrusion detection system for unmanned aerial vehicle networks. The system uses deep reinforcement learning belief networks to detect intrusions. Deep reinforcement learning models combine the reinforcement learning model and deep neural networks, which allows the reinforcement learning agent to be explored separately. The scheme first removes unwanted data from the network data and converts it to a compatible format, and then uses a deep belief network to detect intrusions into the UAV network. The special feature of this solution is the optimization of the deep learning technique using the black widow optimization technique, which greatly improves the accuracy of intrusion detection. This scheme has been experimentally demonstrated to achieve an accuracy rate of 98.9% and a recall rate of 99.3%. Sudar et al. [51] identified malware in software-defined networks using traffic features. This scheme uses traffic data collected from traffic table entries as features to identify network attacks using support vector machines and decision trees as detectors. The warning controller removes the specific traffic from the flow table if network traffic is identified as attack data. The authors tested the performance of this scheme using the KDD99 dataset, and the experimental results show that the support vector machine performs better than the decision tree model in a simulated environment. Overall, the dynamic network traffic-based detection methods focus on malware for DDoS attacks and the samples’ network behavior rather than on the samples’ behavior within the system.

### 2.3. Hybrid Feature-Based Methods

Cozzi et al. [52] proposed a Linux malware-based analysis pipeline using metadata of files, static features, and dynamic features. In this paper, 10,548 samples from ten different CPU architectures were analyzed and used as a baseline to summarize various malware behaviors in Linux. Carrillo-Mondéjar et al. [8] developed a data-driven approach for cross-architecture relations among IoT malware. This method used hybrid features to group unknown malware with known malware using similarity computation. Ban et al. [53] suggested a hybrid analysis approach to analyze IoT malware. Specifically, the authors used a support vector machine (SVM) to classify malware using opcodes as static features and application programming interface (API) features invoked at the execution time of each sample as dynamic features.

Nevertheless, all three hybrid analysis methods are similar to the VMI-based detection methods, as the number of dynamic features used is related to the number of CPU architectures covered by their sample set.

### 2.4. Vmi Detection-Based Features

Tian et al. [10] proposed a malware detection scheme for virtualized environments based on the X86 architecture called MDCHD. This method first collected the information flow of the execution control of the target program using the Intel Processor Trace (IPT) mechanism. Subsequently, the control flow information was converted into color images as features to identify malware using a deep learning approach based on a convolutional neural network (CNN) model. Chen et al. [11] designed and implemented a virtual machine-based malware behavior analysis system for the IoT, which targets malware compiled for the x86-64 and ARM architectures. The sequence of API calls inside the virtual machine was obtained through VMI, transformed into a family behavior graph, and input into a hidden Markov model for classification. Li et al. [54] designed and implemented a security system based on a Virtual Machine Monitor (VMM) named VSyscall. This scheme interrupts instructions to obtain system calls from processes in a virtual machine and correlates the two system calls via the instruction pointer. Thus, VSyscall monitors the processes in the virtual machine without making any changes to the virtual machine. Mishra et al. [55] introduced a cloud-based malware scheme built on the X86 architecture. The program generates a log containing the process list and their details in the virtual machine using strace. The log of the system calls within the virtual machine is obtained and processed in the VMM through the Nitro library. After that, the two system call logs are compared to determine whether the virtual machine was infected with malware. Moreover, an integrated learning model was used as a classifier, using system calls as features, to determine whether running processes inside the virtual machine belong to malware. Cheng et al. [56] presented an IoT analytics platform with a web interface called the ELF Analyzer. This platform’s key component is an IoT sandbox that uses QEMU to emulate various CPU architectures. Additionally, this scheme relies on strace to generate the corresponding system calls and traffic logs for dynamic analysis after running the executable and linkable format (ELF) files uploaded by the users in the sandbox. Jeon et al. [57] introduced a dynamic analysis strategy for IoT malware detection (DAIMD), which, after extracting behaviors related to memory, network, virtual file system, processes, and system calls in an ARM-based virtual machine, converts the extracted behavioral data to images. Then a CNN is used for training and classification.

## 3. Methodology

### 3.1. Overview

This section describes the developed cross-architectural dynamic analysis, entitled MDABP. Figure 1 reveals that MDABP analyzes the given dataset in four steps: PaaS model building, feature extraction, feature selection, and classification model building. In the PaaS model building step, we build two virtual machines in the host machine to ensure that the samples of the corresponding CPU architecture can be executed. In the feature extraction step, we input the samples sequentially into a virtual machine to execute and then record the log of system calls generated by the virtual machine in the host. Each log’s number of system calls is counted through a Python script and converted into a system call vector. In the feature selection step, the dimensionality of the system call vectors is reduced after we remove the system calls with a total number of zero, and the system call vectors of all samples are combined into two feature datasets. Finally, we input two feature datasets into the K Nearest Neighbor (KNN) model for training and classification. Next, we describe each step in detail.

### 3.2. Paas Model Building

We first required the PaaS model to set up the experimental environment for the following experiments. Platform as a Service (PaaS) [58] provides users with support platforms for developing, running, and operating application software through the internet. Similar to personal computer software development, programmers may use development tools to develop and deploy application software on a computer with a Windows or Linux operating system. PaaS has the following operations:

First, virtualization is enabled in the Basic Input Output System (BIOS), and then a desktop version of Ubuntu 21.10 with the AMD64 architecture is installed as the host system. Once installed, the KVM in the Ubuntu system is loaded as a kernel module, turning the Linux kernel into a VMM. After that, QEMU, libvirt, and the virt-manager are installed. Libvirt is a virtual machine management tool, while virt-manager is a graphical management software for managing virtual machines through libvirt. Once libvirt is installed, the system will employ a script to create a network address translation (NAT) network and connect virtual machines created with virt-manager to that network. Next, we create two virtual machines, an i386-based virtual machine and an ARM-based one, using virt-manager. The two VMs operate in Debian, where the ARM-based VM is complemented by the corresponding unified extensible firmware interface (UEFI). After installation, we install a strace on each virtual machine and the host OS to record system calls. Finally, we save the current state of newly created VMs by creating snapshots.

### 3.3. Feature Extraction

Once the experimental environment has been constructed, we pass samples into the VM and collect the system calls generated by the VM on the host machine.

Before presenting the specific steps, we first state the two key points of this experiment. In a virtual machine, we run one sample, during which the other virtual machine is switched off. Besides, the experiments use MobaXterm to connect to the host and virtual machine separately via Secure Shell (SSH) and use this connection to send samples to the virtual machine. MobaXterm is an enhanced client with an integrated Unix command set toolbox that helps us connect to and operate Linux servers in Windows operating systems. When using MobaXterm to connect to a virtual machine, we set the host’s IP address as the gateway address and use the IP address of the virtual machine as the destination address to jump the network address. These two points ensure that the system calls we collect are uncontaminated and usable. After introducing the key points of the experiment, we next describe the specific steps of feature extraction. The specific feature extraction steps are as follows:

First, we start a VM on the host and obtain its process ID using the shell command. Next, we connect the VM and send samples to it. For a stable CPU utilization of about 3%, the strace command is executed on the host to keep track of all system calls generated by the VM process within one minute by the process ID and save them to a txt file. Once the strace command is active on the host, we quickly run the sample on the VM with the strace command, and by analyzing the result of the strace command, the VM’s sample execution state is determined. If the sample does not stop after one minute of execution, the strace command is first stopped on the host, and a snapshot is taken to restore the virtual machine to its state before receiving the sample.

Python calculates the number of system calls in each txt file using Ubuntu 21.10 Desktop Version system calls [59] and saves the results as a dictionary in a JavaScript Object Notation (JSON) file, where the key is the system call name. The value is the number of times this system call was made. Then, by adopting [59], we construct a vector of the number of system calls in each JSON file based on the corresponding system call names. System call counts with no records in the JSON file are recorded as zero. For each system call vector, the behavioral features of a sample are recorded after it has been automatically mapped and normalized. 

### 3.4. Feature Selection

This step aims to reduce the dimensionality of the system call vector obtained in the previous step, thereby reducing the computational burden during classification.

After assembling the system call vector of all samples into a Comma-Separated Values (CSV) file, we compute the total number for each system call. System calls with zero entries are deleted. At this stage, we chose 146 system calls out of 404 as classification features. Finally, we group all the classification features of the samples into two feature datasets based on their CPU architectures and name them ARM-based and X86-32-based datasets. 

### 3.5. Classification Model Building

The two feature datasets obtained in the previous step are used to train and evaluate the classification model.

The k nearest neighbors (KNN) model is selected as the MDABP classification model, which determines the classification of unknown samples by calculating the distance between them and all available samples. We chose KNN as the classifier because it has no training process compared to other classification models and the fit is adjustable, so the algorithm is simpler. We use 146 system calls as features for the KNN model classification to discriminate benign samples from malicious samples. The training-testing ratio is 70% and 30% of the dataset. During testing, we evaluate the average accuracy of the KNN model under different parameters using 10-fold cross-validation. 

## 4. Experimental Evaluation

This section conducts several experiments to evaluate the detection effectiveness and performance of MDABP. First, the data set and the experimental equipment are introduced. Then, the effectiveness of MDABP is evaluated by testing its accuracy on a single feature dataset. Next, the dataset based on the single CPU architecture is used as the training set and the other CPU architecture as the testing set to evaluate the MDABP’s ability to detect cross-architecture unknown samples. Finally, after merging the two feature data sets, we used 70% of the training set and 30% of the test set to evaluate the MDABP’s performance in detecting cross-platform mixed samples.

### 4.1. Dataset and Experiment Setting

We collected 1277 malicious samples in ELF format from VirusTotal [60]. In this experiment, each sample is recognized as malware solely based on its corresponding JSON file, without distinguishing between malware families or types. Next, we collected 442 benign samples from the OSs of two VMs [20]. Table 1 reports the number of examples of different CPU architectures.

All experiments are conducted on a laptop with the Windows 10 operating system, an AMD Ryzen 7 4800H CPU (2.9 GHz), and 16 GB of RAM. Moreover, we create a new VM in VMware Workstation 16.2 using Ubuntu 21.10 as the host OS. The virtual machine created in VMware Workstation 16.2 is linked to the Windows 10 network via NAT.

### 4.2. Evaluation Metrics

The MDABP’s performance is evaluated using the following standard evaluation metrics: 

Accuracy is the percentage of correctly classified test samples:(1)$$\begin{array}{c} \mathrm{Accuracy}=\frac{\mathrm{TP}+\mathrm{TN}}{n} \end{array}$$

Precision is the probability that the predicted positives are correctly classified:(2)$$\mathrm{Precision}=\frac{\mathrm{TP}}{\mathrm{TP}+\mathrm{FP}}.$$

Recall is the probability of correctly classifying a sample in a given class:(3)$$\mathrm{Recall}=\frac{\mathrm{TP}}{\mathrm{TP}+\mathrm{FN}}.$$

F1-measure is the weighted average of precision and recall:(4)$$F1\mathrm{measure}=\frac{2*\left(\mathrm{Recall}*\mathrm{Precision}\right)}{\mathrm{Recall}+\mathrm{Precision}}.$$
where True Positive (TP) denotes the samples correctly classified as positive, False Positive (FP) are the samples incorrectly classified as positive, True Negative (TN) are the samples correctly classified as negative, and False Negative (FN) are the samples incorrectly classified as positives.

### 4.3. Evaluation of Performance on a Single Architecture Sample Set

In the first set of experiments, we evaluate MDABP’s performance using a single architectural dataset, validating the concept of employing what the system calls as classification features. 

The first trial evaluates the ARM-based dataset, which comprises 723 malicious samples and 223 benign ones. The average training and testing accuracy is illustrated in Figure 2. After comparing the average accuracy of the KNN model with 10-fold cross-validation at various values of k, we set k = 23 because KNN achieves the highest mean accuracy of 98.34% and a 99.86% recall rate.

Next, we retested MDABP on the X86-32-based dataset, which contains 554 malicious samples and 219 benign samples. The corresponding average training and testing accuracy are presented in Figure 3. After trial and error, for k = 3, the KNN model achieves the highest average accuracy of 97.62% and the highest recall rate of 99.34%.

Existing cross-architecture dynamic analysis lacks a method that treats system calls as the only feature type. Therefore, we use two static analysis methods based on a single architecture [5,6] and compare them against MDABP. Specifically, Hu et al. [5] proposed an ARM architecture-based IoT malware detection model that employed opcodes as features for classification. The authors constructed a hierarchical transformer model (HTM) to classify samples by exploiting the internal hierarchical structure of functions in malware. The experimental results revealed that HTM achieved 94.67% accuracy and a 99.12% recall rate in malware classification. Moreover, Niu et al. [6] developed a malware detection method based on the X86 architecture, which relies on fused features. They used information entropy and CFG to represent the global correlation of opcode features and detect malware using the Extreme Gradient Boosting (XGBoost) model. They obtained an accuracy of 94.6% and a 94.7% recall rate. 

Table 2 reports the accuracy of MDABP compared to existing research. Regarding accuracy in detecting samples based on a single CPU architecture, MDABP is better than existing malware classification models, demonstrating that the system calls generated by the virtual machine on the host machine can be efficient classification features.

### 4.4. Evaluation of Performance on Cross-Architectural Samples

Next, we evaluated the MDABP’s performance using a cross-architecture sample. Specifically, first, we evaluated MDABP’s accuracy using the ARM-based dataset as the training set and the X86-32-based dataset as the testing set. After that, the accuracy of MDABP was re-evaluated using the X86-32-based dataset as the training set and the ARM-based dataset as the testing set. In this way, we assess the cross-architectural ability of MDABP to detect unknown samples.

The evaluation results on datasets using different CPU architectures are reported in Table 3. When the ARM-based dataset is used as a training set and the X86-32-based dataset is used for testing, the accuracy of MDABP is 83.5%. In the opposite case, the accuracy of MDABP is 93.9%. The reason is that the ARM-based dataset has features with much larger values than the features in the X86-32-based dataset.

### 4.5. Evaluation of Mixed Samples of Cross-Architecture

The next set of experiments mixes the datasets into a single dataset and then evaluates the performance of MDABP using this mixed dataset. Following the same experimental evaluation as in the previous trials, the accuracy of MDABP is 97.18% and the recall rate is 99.01% for k = 13.

Finally, we compare the performance of MDABP against the existing malware classification methods. Since existing dynamic cross-architecture analysis methods that use a single set of system calls as features are not available, we compare MDABP with dynamic analysis methods based on the characteristics of network traffic [45,46,47]. Specifically, Kumar et al. [45] proposed a distributed modularization scheme based on a network traffic database that involves three machine learning models as classification models. After experimental validation, the authors demonstrated that the KNN model has the highest accuracy of 94.44% and a 100% recall rate. Palla et al. [46] detected Mirai viruses in IoT devices using an artificial neural network (ANN) to highlight packets in network traffic. Their method was experimentally verified to attain an accuracy rate of 92.9% and a recall rate of 99%. Bendiab et al. [47] introduced a malware detection method based on packet-level network traffic that uses a residual neural network (ResNet50) as the detection model, achieving 94.5% detection accuracy and a 94.02% recall rate. The comparison results are reported in Table 4, highlighting that the classification accuracy of the proposed cross-architecture MDABP model achieves higher accuracy than the existing malware classification models based on network traffic.

## 5. Conclusions

The Internet of Things is widely used in various aspects of everyday human activity due to the development of internet technology. While providing significant economic benefits, the IoT poses significant security threats. Currently, the available static cross-architecture analysis methods require complex feature extraction and discarding samples that cannot be decompiled. Accordingly, a dynamic cross-architecture analysis must consider the diversity of system calls generated by samples from different CPU architectures. To address these issues, we present MDABP, a cross-architecture dynamic detection approach for IoT malware based on the KVM principle and the PaaS model. The developed scheme first creates two virtual machines with X86-32 and ARM CPU architectures in the desktop version of the Ubuntu 21.10 operating system on the X86-64 architecture. The system calls generated by the VM in the host OS are then recorded while executing one sample at a time in a single, running virtual machine. Finally, system calls are used as features, and the KNN model is used as a classification model to detect IoT malware based on different CPU architectures. To our knowledge, MDABP is the first dynamic cross-architecture analysis method that explicitly relies on system calls in a single set as features.

In this study, we evaluate the accuracy of MDABP on our dataset, achieving 98.34% accuracy and a 99.86% recall rate in ARM-based malware detection. The detection accuracy for X86-32-based malware is 97.62%, and the recall rate is 99.34%. Moreover, the accuracy of MDABP is 97.18%, and the recall rate is 99.01% in the mixed architecture samples scenario. Overall, the experimental results prove that our approach effectively detects cross-architecture IoT malware.

## 6. Discussion and Future Work

The proposed MDABP is a cross-architectural dynamic analysis technique that detects malware by capturing the behavioral features of the VM within the host at the time of sample runtime. MDABP opposes static analysis techniques that obtain opcodes and system calls within files through disassembly techniques. Thus, MDABP can obtain dynamic features even if obfuscation techniques process the samples. It is essential to highlight that as long as the VM is active on the host, the strace command can be used in the host to track and record the system calls generated by the VM. Hence, even if the malware cannot be run, we can still record the system calls generated in the host using the VM where the sample is and mark the system call log as malicious. In addition, MDABP detects samples compiled based on different architectures using system calls in specific operating systems. MDABP significantly reduces the researcher’s workload in feature selection compared to the “unique feature” approach. We next describe the shortcomings of MDABP and suggest ways to improve it.

First, MDABP does not make all samples run smoothly, probably because the operating system is unsuitable or the corresponding files are missing. Therefore, in future work, we will study the ELF files to determine how to make each sample perform smoothly.

The second disadvantage of MDABP is the complex steps required to create a virtual machine. This makes MDABP difficult to use, so we will investigate how to simplify MDABP. MDABP is a method to detect cross-architecture IoT malware on personal computers, which requires high computer performance because it needs to run virtual machines. We conduct one-minute tests in fixed cycles to ensure that sufficient dynamic features are collected. As a result, the analysis process of MDABP is more time-consuming than the lightweight detection models targeted for deployment on IoT devices. In the future, we will simplify MDABP to reduce the time consumed by the analysis process.

MDABP utilizes the quarantine feature of virtualization technology to restrict the harmfulness of malware. Some samples may have identification mechanisms to evade detection or break isolation restrictions to infect hosts. We plan to address this issue through static analysis. Therefore, future work will investigate an MDABP-based hybrid analysis technique that combines dynamic and static analysis techniques to achieve higher detection accuracy.

## Figures and Tables

**Figure 1 sensors-23-03060-f001:**
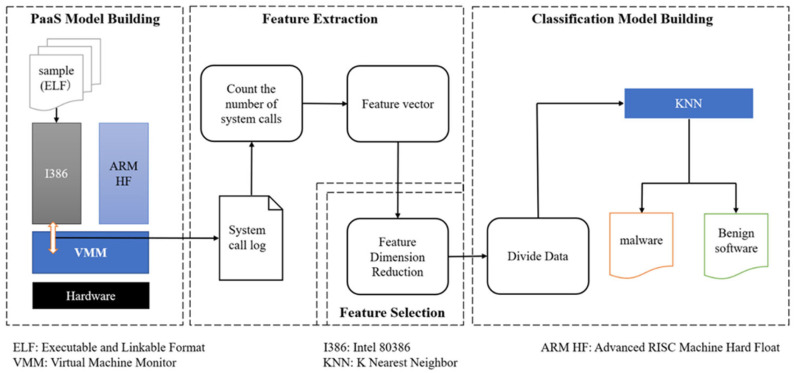
System model.

**Figure 2 sensors-23-03060-f002:**
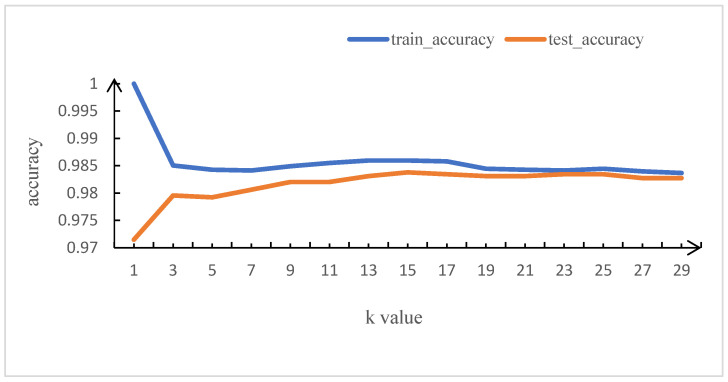
Average training and testing accuracy based on the ARM dataset.

**Figure 3 sensors-23-03060-f003:**
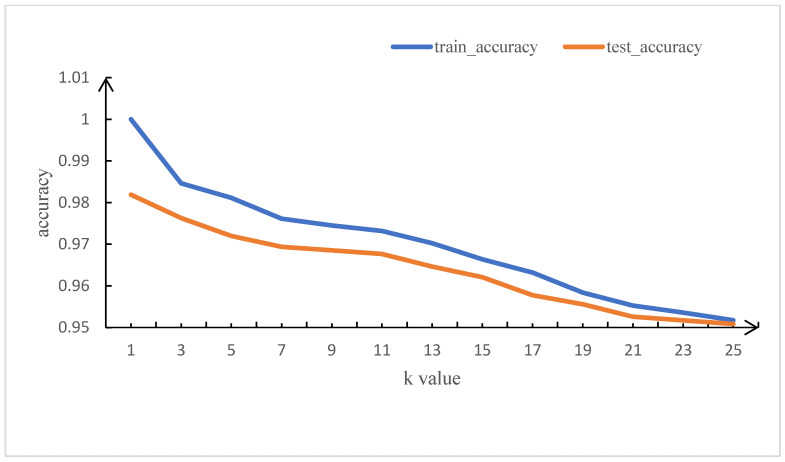
Average training and testing accuracy based on the X86-32 dataset.

**Table 1 sensors-23-03060-t001:** Sample distribution for each CPU architecture.

	ARM	X86-32	Total
malware	723	554	1277
benign	223	219	442

**Table 2 sensors-23-03060-t002:** Single CPU architecture performance.

Algorithm	Architecture	Detection Model	Accuracy (%)	Recall (%)
Hu et al. [5]	ARM	HTM	94.67	99.12
MDABP	ARM	KNN	98.34	99.86
Niu et al. [6]	X86	XGboost	94.60	94.70
MDABP	X86-32	KNN	97.62	99.34

**Table 3 sensors-23-03060-t003:** Multi-type CPU architecture performance.

IoT Dataset		Accuracy (%)	Recall (%)	F1 (%)
Trained by	Evaluated by			
ARM	X86-32	83.50	100	89.71
X86-32	ARM	93.90	94.32	95.98

**Table 4 sensors-23-03060-t004:** Comparison of MDABP against existing malware classification models.

Algorithm	Feature	Detection Model	Accuracy (%)	Recall (%)	F1(%)
Kumar et al. [45]	network traffic	KNN	94.44	100	96.00
Palla et al. [46]	network traffic	ANN	92.90	99.00	95.00
Bendiab et al. [47]	network traffic	ResNet50	94.50	94.02	94.90
MDABP	system call	KNN	97.18	99.01	98.08

## Data Availability

The data presented in this study are available on request from the corresponding author.

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
