# Peer review of "MDABP: A Novel Approach to Detect Cross-Architecture IoT Malware Based on PaaS"

_sensors, 2023, doi:10.3390/s23063060_

Round 1
Reviewer 1 Report
1. While the abstract has aimed to provide a comprehensive overview of the main contribution, there is a need to be revised so that the general reader can grasp the main idea/topic of the draft and the main contribution.
2. For section 3.3, the authors mentioned “MobaXterm,” but there is no description, so first describe it. Moreover, the authors also said that separation between VM is based on Secure Shell, but there is no description of how the secure connection is established b/w VM during message passing.
3. In section 3.4, which feature selection technique is used?
4. The authors used the KNN classification model; however, the many latest models used SVM and the Decision tree model, so the authors should describe why the proposed KNN-based model is more efficient.
5. Before you conclude the manuscript in the conclusion section, I would like to see a discussion chapter. This discussion chapter should describe the implications for research and practice of your work and honestly describe your work's limitations (e.g., whether certain trust assumptions are still necessary for it to work). I think this chapter helps the readers a lot to put your work into the broader context/developments in the field.
6. What is next? What are future directions? Authors should include some future work of their model.
Author Response
Reviewer 1
Comments and Suggestions for Authors:
- While the abstract has aimed to provide a comprehensive overview of the main contribution, there is a need to be revised so that the general reader can grasp the main idea/topic of the draft and the main contribution.
Reply: I have modified some sentences to clarify the contribution of this article. Those sentences are shown below and highlighted in yellow:
As IoT devices are increasing rapidly, their security must classify malicious software accurately. However, current IoT malware classification methods cannot detect cross-architecture IoT malware using system calls in a particular operating system as the only class of dynamic features.
- For section 3.3, the authors mentioned “MobaXterm,” but there is no description, so first describe it. Moreover, the authors also said that separation between VM is based on Secure Shell, but there is no description of how the secure connection is established b/w VM during message passing.
Reply: I have added some sentences as you suggested, highlighted in yellow and presented below.
Besides, the experiments use MobaXterm to connect to the host and virtual machine separately via Secure Shell (SSH) and use this connection to send samples to the virtual machine. MobaXterm is an enhanced client with an integrated Unix command set toolbox that helps us connect and operate Linux servers in Windows operating systems. When using MobaXterm to connect to a virtual machine, we set the host's IP address as the gateway address and use the IP address of the virtual machine as the destination address to jump the network address.
- In section 3.4, which feature selection technique is used?
Reply: We did not use an algorithm to select features, and we just removed the system calls with zero total numbers by manual counting and used the remaining system calls as classification features.
- The authors used the KNN classification model; however, the many latest models used SVM and the Decision tree model, so the authors should describe why the proposed KNN-based model is more efficient.
Reply: I have added the explanation as you suggested. The sentences explained have been marked in yellow.
We chose KNN as the classifier because it has no training process compared to other classification models, and the fit is adjustable, so the algorithm is simpler.
- Before you conclude the manuscript in the conclusion section, I would like to see a discussion chapter. This discussion chapter should describe the implications for research and practice of your work and honestly describe your work's limitations (e.g., whether certain trust assumptions are still necessary for it to work). I think this chapter helps the readers a lot to put your work into the broader context/developments in the field.
Reply: I have explained the limitations of this study as you requested.
First, MDABP does not make all samples run smoothly, probably because the operating system is unsuitable or the corresponding files are missing.
The second disadvantage of MDABP is the complex steps required to create a virtual machine.
MDABP is a method to detect cross-architecture IoT malware in personal computers, which requires high computer performance because it needs to run on virtual machines. We conduct one-minute tests in fixed cycles to ensure that sufficient dynamic features are collected. As a result, the analysis process of MDABP is more time-consuming than the lightweight detection models targeted for deployment on IoT devices.
MDABP utilizes the quarantine feature of virtualization technology to restrict the harmfulness of malware. Some samples may have identification mechanisms to evade detection or break isolation restrictions to infect hosts.
- What is next? What are future directions? Authors should include some future work of their model.
Reply: Based on the abovementioned limitations, I have added the following statements to the text to illustrate future plans.
Therefore, in future work, we will study the ELF files to determine how to make each sample perform smoothly.
This makes MDABP difficult to use, so we will investigate how to simplify MDABP in future work.
In the future, we will simplify MDABP to reduce the time consumed by the analysis process.
We plan to address this issue through static analysis. Therefore, future work will investigate an MDABP-based hybrid analysis technique that combines dynamic and static analysis techniques to achieve higher detection accuracy.

Reviewer 2 Report
The author introduced A Novel Approach to detect Cross-Architecture IoT Malware Based on PaaS. However, I have some suggestions like
1. Figures 2 and 3 quality is not good. The author should improve the way of presentation of outputs in terms of figures.
2. Tables 2, 3, and 4 should be represented in the form of a graphical representation like a graph.
3. The Introduction part is weak, the author should discuss the following article in the introduction or related works part. (https://doi.org/10.32604/cmc.2022.020066, 10.1109/ICCCI50826.2021.9402517, https://doi.org/10.1016/B978-0-12-819511-6.00006-6)
4. Timing analysis is missing in the article.
5. The security analysis is also missing in the article. If necessary, please include that also.
Author Response
Reviewer 2
Comments and Suggestions for Authors:
The author introduced A Novel Approach to detect Cross-Architecture IoT Malware Based on PaaS. However, I have some suggestions like
- Figures 2 and 3 quality is not good. The author should improve the way of presentation of outputs in terms of figures.
Reply: I have changed the picture. Please refer to the updated version.
- Tables 2, 3, and 4 should be represented in the form of a graphical representation like a graph.
Replay: I have replaced the form with a picture, as requested.
- The Introduction part is weak, the author should discuss the following article in the introduction or related works part. (https://doi.org/10.32604/cmc.2022.020066, 1109/ICCCI50826.2021.9402517, https://doi.org/10.1016/B978-0-12-819511-6.00006-6)
Reply: I have added the first two articles you recommended to this article following the related work. I did not insert the third article you recommended because I could not read it, as my university does not purchase the journal to which the article belongs.
Praveena et al. [51] proposed an intrusion detection system for unmanned aerial vehicle networks. The system uses deep reinforcement learning belief networks to detect intrusions. Deep reinforcement learning models combine the reinforcement learning model and deep neural network, which allows the reinforcement learning agent to be explored separately. The scheme first removes unwanted data from the network data and converts it to a compatible format, and then uses a deep belief network to detect intrusions into the UAV network. The special feature of this solution is the optimization of the deep learning technique using the black widow optimization technique, which greatly improves the accuracy of intrusion detection. This scheme has been experimentally demonstrated to achieve an accuracy rate of 98.9% and a recall rate of 99.3%. Sudar et al. [52] identified malware in software-defined networks using traffic features. This scheme uses traffic data collected from traffic table entries as features to identify network attacks using support vector machines and decision trees as detectors. The warning controller removes the specific traffic from the flow table if network traffic is identified as attack data. The authors tested the performance of this scheme using the KDD99 dataset, and the experimental results show that the support vector machine performs better than the decision tree model in a simulated environment.
- Timing analysis is missing in the article.
Reply: I have compared the algorithm running times of several methods. Additionally, the proposed method is testing samples with a fixed period of one minute. Therefore, in the last section of this paper, it is demonstrated that this method is more time-consuming than the existing methods and an improvement plan for this weakness is proposed.
MDABP is a method to detect cross-architecture IoT malware in personal computers, which requires high computer performance because it needs to run virtual machines. We conduct one-minute tests in fixed cycles to ensure that sufficient dynamic features are collected. As a result, the analysis process of MDABP is more time-consuming than the lightweight detection models targeted for deployment on IoT devices. In the future, we will simplify MDABP to reduce the time consumed by the analysis process.
- The security analysis is also missing in the article. If necessary, please include that also.
Reply: As you suggested, I added the analysis in the last section. The added sentences are shown in yellow markings.
MDABP utilizes the quarantine feature of virtualization technology to restrict the harmfulness of malware. Some samples may have identification mechanisms to evade detection or break isolation restrictions to infect hosts. We plan to address this issue through static analysis. Therefore, future work will investigate an MDABP-based hybrid analysis technique that combines dynamic and static analysis techniques to achieve higher detection accuracy.

Round 2
Reviewer 2 Report
After the revisions, the paper quality improved a lot. The authors fulfill all my comments. Now, I recommend this paper in its current form.